# Three-dimensional monolithic integration in flexible printed organic transistors

Jimin Kwon [1], Yasunori Takeda [2], Rei Shiwaku[2], Shizuo Tokito[2], Kilwon Cho[3] & Sungjune Jung [1,3]

Direct printing of thin-film transistors has enormous potential for ubiquitous and lightweight wearable electronic applications. However, advances in printed integrated circuits remain very rare. Here we present a three-dimensional (3D) integration approach to achieve technology scaling in printed transistor density, analogous to Moore's law driven by lithography, as well as enhancing device performance. To provide a proof of principle for the approach, we demonstrate the scalable 3D integration of dual-gate organic transistors on plastic foil by printing with high yield, uniformity, and year-long stability. In addition, the 3D stacking of three complementary transistors enables us to propose a programmable 3D logic array as a new route to design printed flexible digital circuitry essential for the emerging applications. The 3D monolithic integration strategy demonstrated here is applicable to other emerging printable materials, such as carbon nanotubes, oxide semiconductors and 2D semiconducting materials.

[1] Department of Creative IT Engineering, Pohang University of Science and Technology (POSTECH), 77 Cheongam-Ro, Nam-Gu, Pohang 37673, Republic of Korea. [2] Research Center for Organic Electronics (ROEL), Graduate School of Science and Engineering, Yamagata University, 4-3-16 Jonan, Yonezawa, Yamagata 992-8510, Japan. [3] Department of Chemical Engineering, Pohang University of Science and Technology (POSTECH), 77 Cheongam-Ro, Nam-Gu, Pohang 37673, Republic of Korea. Correspondence and requests for materials should be addressed to S.T. (email: tokito@yz.yamagata-u.ac.jp) or to K.C. (email: kwcho@postech.ac.kr) or to S.J. (email: sjjung@postech.ac.kr)

D irect printing of thin-film transistors on plastic foil has been shown to be a highly promising technology for fabricating ubiquitous and lightweight wearable electronic applications[1]. The potential of this additive manufacturing technique for low-cost mass production, rapid prototyping of small series, and low-temperature fabrication on flexible substrates renders it an attractive alternative to conventional fabrication methods for electronics[2,3]. Solution-processable semiconductor materials in the forms of organic compounds, graphene, carbon nanotubes, and oxides have been developed to be compatible with current printing techniques[4–7]. Among these materials, organic semiconductor (OSC) small molecules or polymers have been distinguished from other candidates as softer materials and have been successfully demonstrated in many flexible printed devices, such as transistors in an array[8], nonvolatile memory cells[9], electrochemical transistor-based biosensors for biological research[10], display light-emitting diodes[11], and solar cells for energy harvesting[12].

Nevertheless, advances in printed organic integrated circuits (ICs) remain very rare. The primary challenge is the lack of technology scaling in printed transistors. The scaling capability of conventional silicon transistors, i.e., Moore's law, which calls for the transistor density to be doubled every 18 months, has made ICs an excellent example of a general purpose technology[13]. Although organic transistors patterned by photolithography or shadow mask evaporation have been applied to microprocessors and smart tags[14,15], most functional organic ICs fabricated by printing have less than a hundred transistors[16–21]. The progress in printed organic ICs has been slow because the low resolution (typically 10–100 μm) and large feature size (typically 100–200 μm) of current printing techniques have limited the transistor counts in processable areas[22]. Emerging flexible printed IC applications beyond displays are now requiring technology scaling to increase transistor density, which is analogue to the Moore's law.

Another challenge in using flexible printed organic transistors in IC design is their low device performance. In particular, flexible printed organic transistors still suffer from partial depletion of the OSC bulk and low transconductance $g_m$. Partial depletion, which indicates the accumulation of the majority of carriers in the OSC bulk even when the gate-to-source voltage $V_{GS}$ is zero, leads to undesirable effects: decreased intrinsic gain, static noise margin, and drain current $I_D$ on-off ratio and increased subthreshold

swing SS (dec/mV) and standby power[23]. Moreover, low $g_m$ is mainly attributed to the low carrier mobilities of OSC materials. Instead of a reduction in the channel length, which is highly limited in printed organic transistors[24], the $g_m$ can be easily elevated by widening the channel width. However, changing the channel geometry necessitates the sacrifice of considerable substrate area.

We previously introduced a three-dimensional (3D) organic transistor structure with a shared gate joining two complementary transistors using drop-on-demand inkjet printing[20]. The study reported an array of the vertically stacked complementary transistors and their logic gates. Building on the approach, here we propose the 3D monolithic integration of flexible printed organic transistors to realize technology scaling and performance enhancement. In addition, we adopt a dual-gate configuration in the 3D transistor-on-transistor structure for the improvement of their electrical characteristics such as SS, transconductance, and $I_D$ on-off ratio. The 3D monolithic integration of dual-gate organic transistors is successfully implemented on a plastic foil with a record density of 60 printed transistors per square centimetre. These transistors exhibits high yield, uniformity, and year-long stability, which prove that this technology is extendable to large-scale printed electronics. By interconnecting those 3D-integrated dual-gate transistors, we finally propose a 3D universal logic gate and its array as a new facile route to design printed digital circuitries that are essential for emerging flexible electronics applications.

## Results

**3D device configuration.** To provide a proof of principle for the 3D monolithic integration, we first demonstrate dual-gate and transistor-on-transistor configurations in printed complementary transistors by vertically stacking functional layers. Starting from a conventional single-bottom-gate transistor on a flexible plastic sheet (Fig. 1a), we fabricated a dual-gate transistor by depositing a top-gate dielectric and inkjet-printing a top-gate Ag electrode on the top (Fig. 1b). This 3D integration of the top and bottom-gate structures enhances the control of the channel by gating on both sides and maximizes the utilization of the transistor. We adopted this dual-gate structure for complementary OSCs: an n-type benzobis(thiadiazole) derivative (TU-3) and a p-type small molecule, dithieno[2,3-d;2′,3′-d′]benzo[1,2-b;4,5-b′]dithiophene

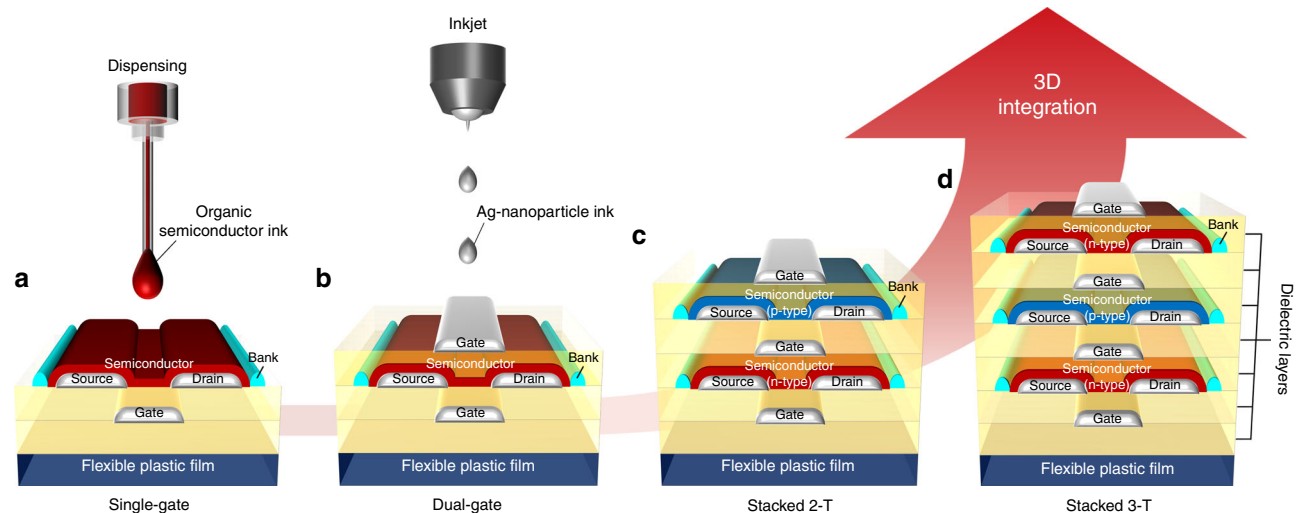

**Fig. 1** 3D monolithic integration in flexible printed transistors. **a**, **b** Printed single-gate and dual-gate organic transistors. **c**, **d** 3D-integrated two (n-/p-type, stacked 2-T) and three (n-/p-/n-type, stacked 3-T) complementary dual-gate organic transistors with shared gate electrodes. All the films were printed except for the Parylene dielectric layers

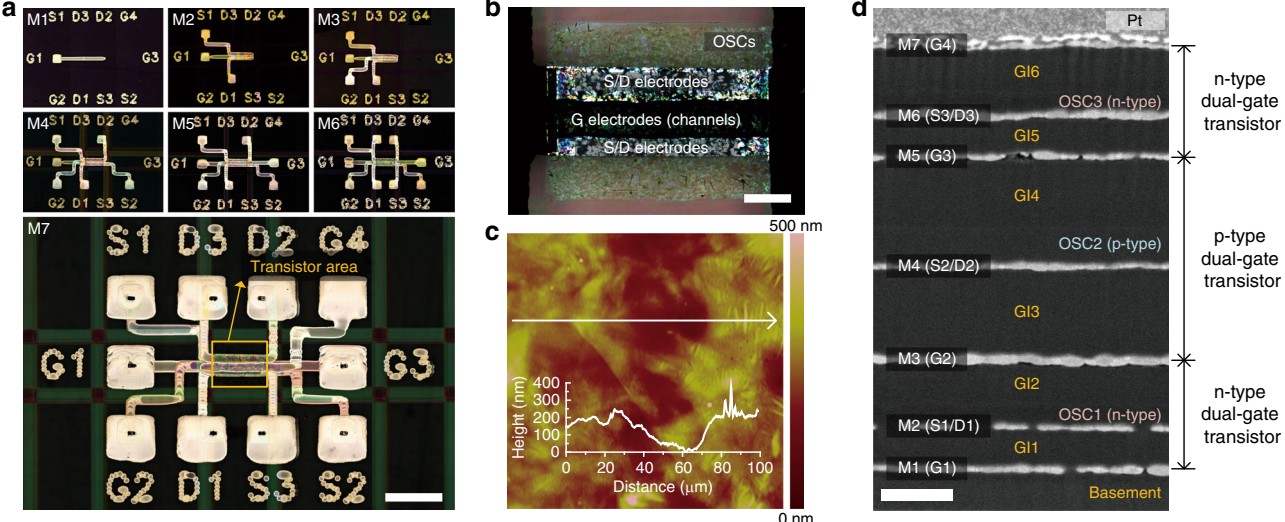

**Fig. 2** Images of a stacked 3-T device. **a** Sequential images of the process of printing seven metal layers (scale bar is 1 mm). **b** A polarized microscopy image of the transistor area (scale bar is 200 μm). **c** A topographic AFM image (scan area is 100 × 100 μm) measured on top of the channel area. The inset graph shows the profile along the cross-section line. **d** Cross-sectional SEM image of the stacked 3-T device (M1-7: metal layers, GI1-6: dielectric insulator layers, OSC1-3: organic semiconductor layers, scale bar is 250 nm, Pt was sputtered on the top prior to the SEM measurement)

(DTBDT-C$_6$) (the OSC chemical structures and the polarized microscopic images are provided in Supplementary Fig. 1). We then implemented a complementary stacked two-transistor (2-T) device by building a dual-gate p-type transistor on top of the n-type transistor (Fig. 1c). The in-between printed gate electrode was shared by the two complementary transistors. The further scalability of the 3D integration was shown by vertically stacking another n-type dual-gate transistor on top of the stacked 2-T device (Fig. 1d). The functional layers of the stacked three-transistor (3-T) device were monolithically integrated via simple repeats of single transistor fabrication without an additional process for 3D integration. These stacked 2-T and 3-T devices are proposed as inverters and universal logic gates, respectively, which are the basic building blocks of fully flexible digital circuits on a plastic sheet.

The stacked 3-T device configuration has 24 functional layers: a plastic substrate, seven conductor layers, three OSC layers, seven Parylene layers (one basement and six gate insulator films), three charge injection self-assembled monolayers, and three bank layers. For all the devices, the source/drain (S/D) and gate (G) electrodes were inkjet-printed, and the semiconducting inks were dispenser-printed within a rectangular hydrophobic fluoropolymer bank that was also made by dispenser-printing. Figure 2a shows optical images of all the metal layers inkjet-printed on each dielectric layer, which can be interconnected on the top floor through laser-drilled via-holes. The availability of the seven metal layers, which is an exceptional number in printed transistors, is key to enabling complex IC designs. A polarized microscopy image of the vertically overlapping semiconducting layers in the transistor area is shown in Fig. 2b. As seen in atomic force microscopy (AFM) data of Fig. 2c, the repeats of the printing processes produced a rough surface morphology in the accumulated polycrystalline semiconductor layers and Ag electrodes (the AFM topology images of the Parylene-coated plastic substrate and individual complementary OSCs can be found in Supplementary Fig. 2). However, conformal chemical vapour deposition of the Parylene dielectrics provided pinhole-free electrical insulation and strong chemical resistance, leading to high yield and uniform device performances. A scanning electron microscopy (SEM) image of the stacked 3-T device confirms that

all the functional layers were conformally deposited as designed without physical damage (Fig. 2d). Detailed information on material preparation, inkjet printing, and device fabrication is described in the Methods section. The entire fabrication process of the stacked 3-T devices is summarized in Supplementary Table 1.

**Characteristics of 3D-integrated dual-gate transistors**. The dual-gate configuration yielded several advantages that helped overcome the chronic problems of conventional single-gate organic transistors. It provides a fully depleted OSC bulk and enhanced $g_m$ due to the effective control of charge transport within the bulk of the semiconductor beyond the highly accumulated region several nanometres in depth[25]. The electrical characteristics of n-type and p-type printed dual-gate transistors, as presented in Fig. 3a, are exceptionally improved compared to those of single-bottom-gate and single-top-gate transistors (schematic cross-section and microscopic images of the transistors are shown in Supplementary Fig. 3). Both gates of the dual-gate transistors were electrically connected for the $V_{GS}$ sweeps. The printed transistor properties were highly uniform and consistent (Supplementary Fig. 4). To demonstrate the compatibility with low-power portable electronics, a relatively low supply voltage $V_{DD}$ of 5 V was used. Surprisingly, for the n-type transistors, the partially depleted OSC behaviour observed in the single-gate transistors was effectively removed in the dual-gate transistors. The turn-on gate voltage shifted from negative to near zero, thereby reducing the subthreshold leakage $I_{OFF}$. In this study, we defined the $I_{OFF}$ as an $I_D$ at zero $V_{GS}$, not as the minimum $I_D$ in a negative bias as often used in the literature, because $I_{OFF}$ is a practical indicator of how much power a transistor wastes in standby when considering actual circuit operations.

In addition, the dual-gate transistors had higher $g_m$ values than single-gate transistors in both complementary OSCs. The improved gating of the fully depleted dual-gate transistors also resulted in the n-type and p-type transistors having a low SS of ≈110 and ≈80 mV dec$^{-1}$, respectively, which are far lower than those of the single-gate transistors (Fig. 3b). These values are competitive with the lowest SS values reported to date for

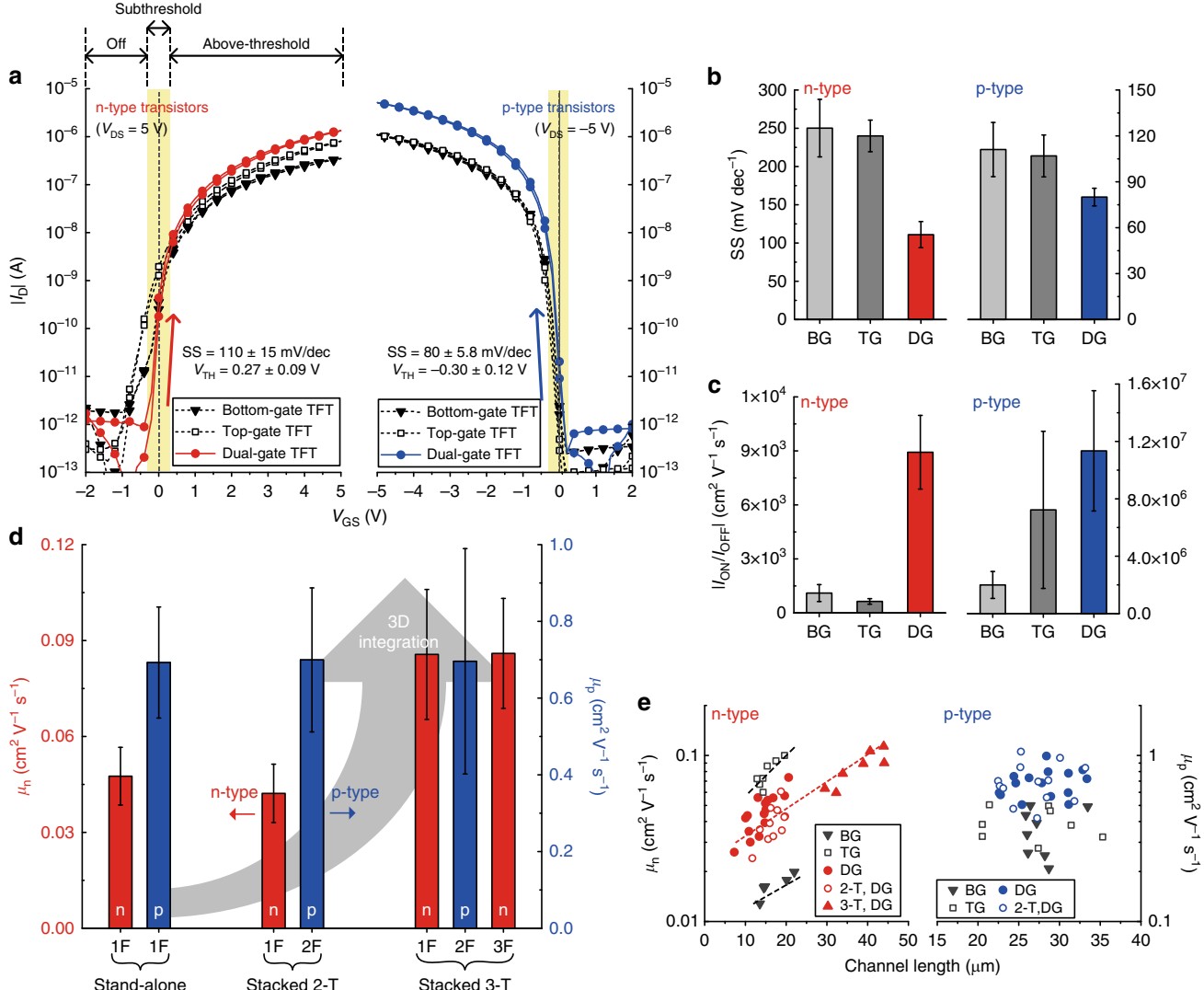

**Fig. 3** Device characteristics. **a** $|I_D|-V_{GS}$ transfer characteristics of the n-type and p-type organic transistors in single-gate (bottom-gate and top-gate) and dual-gate configurations. **b** The subthreshold swings and **c** the $I_D$ on-off ratios. **d** The extracted carrier mobilities in the 3D-integrated complementary organic transistors. **e** The channel length dependency of the extracted carrier mobilities. Error bars represent standard deviation

complementary organic transistors[26]. The low SS enabled a near-zero threshold voltage $V_{TH}$ of 0.27 and $-0.30$ V on average for the n-type and p-type transistors, respectively. The increased $g_m$ and reduced $I_{OFF}$ of the complementary dual-gate transistors led to a considerable improvement in the drain current on-off ratio $I_{ON}/I_{OFF}$ (Fig. 3c). The average $I_{ON}/I_{OFF}$ values of the n-type and p-type dual-gate transistors were increased by ≈710 and ≈470% compared to those of the bottom-gate transistors, respectively, and by ≈1300 and ≈60% compared those of the top-gate transistors. The improvements in the $I_{OFF}$, $g_m$, and SS properties are analogous to the advantages observed in conventional fully depleted silicon-on-insulator multi-gate transistors[27]. Importantly, we first verified that all these benefits of full depletion, which were found in conventional dual-gate silicon transistors, are also observed in printed complementary organic dual-gate transistors on a flexible substrate. The information on the channel lengths $L$ and widths $W$, the extracted carrier mobilities, and the threshold voltages is provided in Supplementary Table 2 with the average values and standard deviations.

Ensuring that the 3D stacking process including consecutive printing and annealing steps does not affect the performance of transistors on the lower floors is important. The carrier mobilities

of the stand-alone and stacked dual-gate transistors, which were extracted from $|I_D|^{1/2}-V_{GS}$ transfer characteristics in the saturation regime, are shown in Fig. 3d. It is noteworthy that there was no significant performance degradation during the 3D stacking process in either complementary dual-gate transistor. In the stacked 3-T device, for example, the carrier mobility of the n-type transistor on the first floor (1F), which went through 14 thermal annealing steps, was almost identical to that of the n-type transistor on the top floor (3F). This result proves that the additive manufacturing process of this work is robust and extendable to further 3D integration of printed transistors.

We observed that the extracted carrier mobilities of the n-type transistors decreased with shortening the channel length, while those of the p-type devices remained unchanged (Fig. 3e). The extracted mobility values are influenced by contact resistance[28]. Thus, we hypothesized that the behaviour of the n-type transistors was dominated by charge injection due to high contact resistance. To support this, we measured and analyzed the contact resistance of the transistors using the transmission line method. As can be seen in Supplementary Fig. 5a, the contact resistance of the n-type dual-gate transistors was 21.4 kΩ cm at $V_{GS} = 5$ V which was ten times higher than that of the p-type

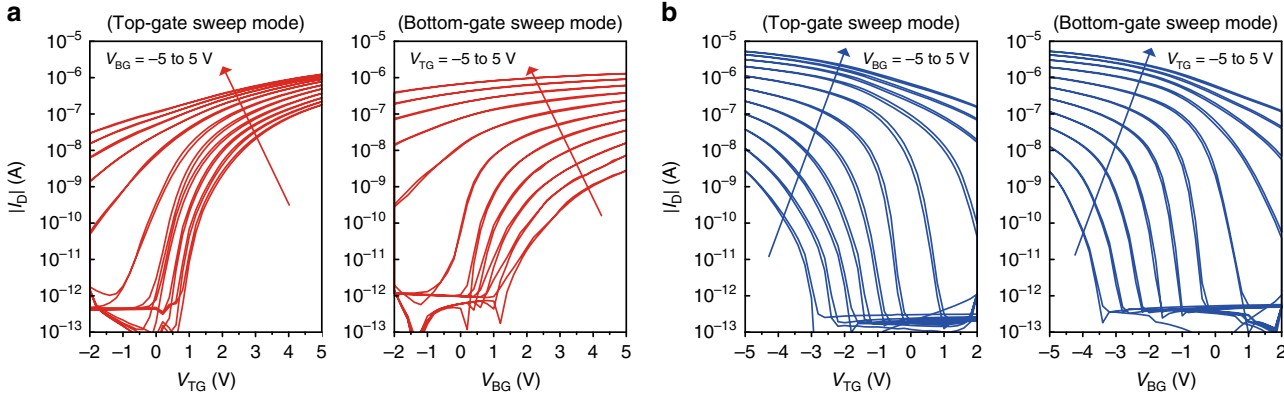

**Fig. 4** Dual-gate complementary transistor operations. **a** Saturation transfer characteristics ($V_{DS} = 5$ V) of the n-type dual-gate transistor in top-gate (left) and bottom-gate (right) sweeping modes ($V_{TG}$: top-gate voltage, $V_{BG}$: bottom-gate voltage). **b** Saturation transfer characteristics ($V_{DS} = -5$ V) of the p-type dual-gate transistor in top-gate (left) and bottom-gate (right) modes

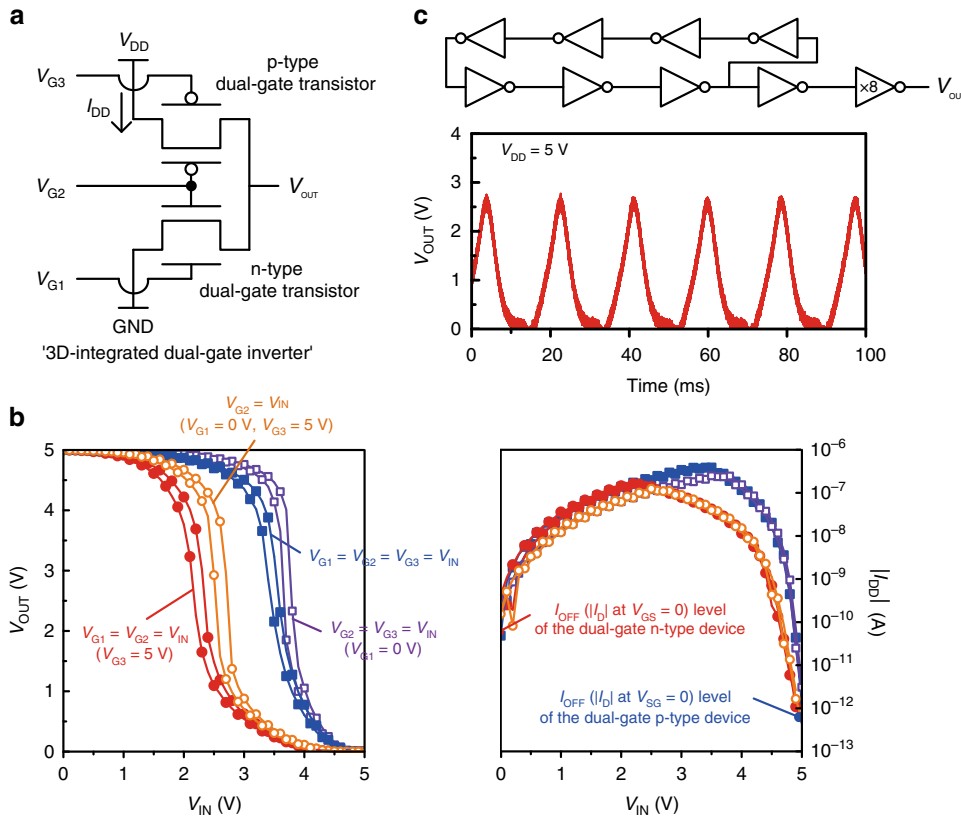

**Fig. 5** Stacked 2-T dual-gate devices. **a** Schematic circuit of the 3D-integrated dual-gate inverter ($V_{G1}$, $V_{G2}$, $V_{G3}$: gate inputs, $I_{DD}$: static current from the source). **b** Voltage transfer characteristics (left) and static currents (right) of the inverter operation according to the combinations of electrical gate connections. **c** Schematic circuit of the ring oscillator which is composed of seven 3D-integrated dual-gate inverters and its oscillation operation. The buffer output stage consists of serially connected two inverters, where the transistor $W$ of the last inverter is designed to be eight times larger than that of the others

dual-gate transistors (2.2 kΩ cm at $V_{GS} = -5$ V). The n-type transistors showed the functional dependence of the contact resistance with the gate bias, which resulted in the improvement of the extracted mobility, whereas the carrier mobilities of the p-type transistors were weakly dependent on the gate bias due to sufficiently low contact resistance (Supplementary Fig. 5b). In addition, the n-type dual-gate devices exhibited highly asymmetrical performance between top-gate and bottom-gate operation modes due to the difference in charge injection efficiency between

the staggered top-gate structure and the bottom-gate structure in the dual-gate transistors (Fig. 4a). In contrast, remarkably symmetrical dual-gate operation was exhibited for the p-type devices because of the high charge ejection efficiency (Fig. 4b).

**3D-integrated flexible circuits.** Next, we implemented a 3D inverter using the stacked 2-T device, the two drain nodes of which are interconnected through a via-hole (the schematic

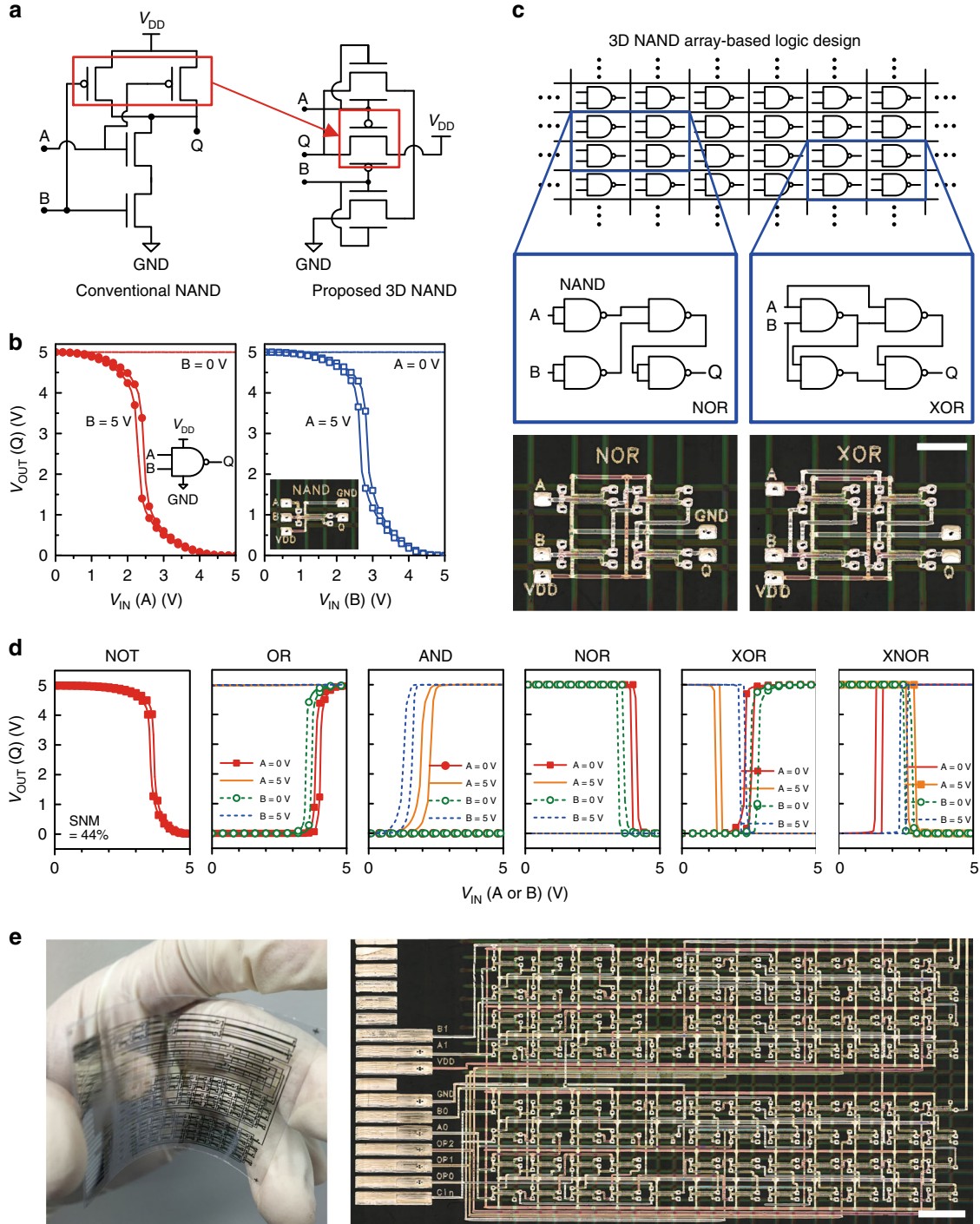

**Fig. 6** 3D NAND digital circuit design based on stacked 3-T dual-gate devices. **a** Schematic circuits of a conventional NAND gate and a proposed 3D NAND gate. **b** DC $V_{OUT}$–$V_{IN}$ characteristics of a 3D NAND gate. **c** 3D NAND array-based logic design (scale bar is 2 mm). **d** DC $V_{IN}$–$V_{OUT}$ characteristics of a 1-input NOT gate and 2-input (A and B) logic gates (OR, AND, NOR, XOR, and XNOR) implemented by interconnecting 3D NANDs. A fixed voltage (0 or 5 V) is applied to one port while a voltage input on the other port is swept. **e** A large-scale flexible logic circuitry implemented by using a 12 × 8 3D NAND gate array (scale bar is 4 mm)

circuit is seen in Fig. 5a). The different combinations of gate connections in the stacked dual-gate complementary transistors were able to maximize static noise margin of the 3D inverter. The inverter output-input voltage characteristics are shown in Fig. 5b according to the four combinations of input gate connections. The inverter switching threshold was optimized to be 50% of $V_{DD}$ when the p-type transistor was operated in bottom-gate mode and the n-type transistor was

operated in dual-gate mode ($V_{G1} = V_{G2} = V_{IN}$, $V_{G3} = 5$ V). Furthermore, we explored the dynamic operation of the 3D-integrated complementary dual-gate transistors by fabricating a 7-stage ring oscillator. The circuit consists of seven inverters for the ring oscillation operation and two inverters for the voltage buffering. The ring oscillator operated with the supply voltage $V_{DD}$ varying between 1 and 15 V. The gate delay was 13 ms at $V_{DD} = 1$ V and 340 μs at $V_{DD} = 15$ V. Figure 5c

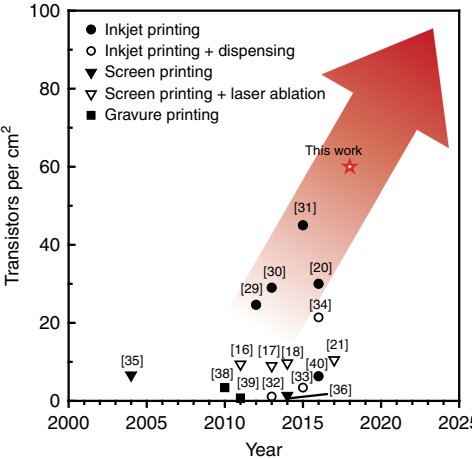

**Fig. 7** Transistor density trend of printed organic circuits fabricated by various printing techniques

depicts the schematic circuit and the oscillation operation at 48.5 Hz when the $V_{DD}$ is 5 V.

We demonstrated a universal logic gate NAND in a 3D configuration by interconnecting transistors in the stacked 3-T device. Unlike a conventional NAND consisting of four complementary transistors, we implemented the NAND gate by using the three vertically stacked dual-gate transistors (Fig. 6a). The independent gate control enabled one p-type transistor in the middle of the stacked 3-T device to operate as two p-type transistors connected in parallel. In the 3D NAND gate, the symmetric bottom-gate and top-gate mode operations of the p-type dual-gate transistor made the output characteristics for the two inputs almost identical (Fig. 6b). To optimize the static noise margin in the 3D NAND gates, we matched the $I_D$ of the dual-gate n-type transistors in dual-gate mode to that of the dual-gate p-type transistor in bottom-gate or top-gate mode by adjusting the thickness of dielectric layers as well as the channel lengths. The two output characteristics exhibited ideal switching operations.

We then introduce a programmable 3D NAND array as a new approach for printed organic digital IC design on a flexible film. The array of the uncommitted gates can be quickly configured for different logic functions by routing them (Fig. 6c). The pitch of $3 \times 2.2$ mm between the 3D NAND gates was determined by considering the space for interconnection routing. The programmable devices allow end users to share a common platform that can be customized with much reduced time and cost. We implemented the first examples of programmable 3D NAND arrays of digital logic circuits, including NOT, AND, OR, NOR, XOR, and XNOR (Fig. 6d). The fabrication process was scaled to produce a large-scale array of 3D NAND gates (Fig. 6e). This work has a record transistor density of approximately 60 transistors per square centimetre for printed ICs when the 3D NAND design is considered to consist of four transistors (Fig. 7)[16,17,20,21,29–40]. Based on this technology, we could fabricate up to ≈2700 programmable transistors on the size of a standard credit card (85.60 × 53.98 mm), which is compatible with the transistor count of the first commercial 4-bit microprocessor.

**Device reliabilities.** Finally, we tested the reliabilities of the 3D-integrated dual-gate complementary transistors fabricated on a flexible substrate under given mechanical, electrical, and environmental stresses. When attached on a curved surface with a bend radius of 8 mm, the dual-gate complementary transistors of

a stacked 3-T device had slight changes in their static transfer characteristics (Fig. 8a). In addition, they exhibited highly stable operation when biased at $V_{GS} = \pm 5$ V for $10^4$ s (Supplementary Fig. 6). Compared to those of the n-type and p-type dual-gate transistors before the bias stress tests, the $I_{ON}$ values of the n-type and p-type dual-gate transistors after the bias stress test were changed by 3 and 5%, respectively, and their $V_{TH}$ shifts were negligible (Fig. 8b). Remarkably, the electrical operations of the printed complementary organic transistors were maintained for 1 year with acceptable performance changes when stored in a dry ambient environment (Fig. 8c). When we observed the corresponding n-type dual-gate transistors 4 months and 1 year after the fabrication, the $\mu_n$ was decreased by 20 and 50%, and the $V_{TH}$ was shifted by 0.4 and 0.7 V, respectively. The p-type transistors exhibited exceptionally small changes in their device parameters: at 1 year after fabrication, the $\mu_p$ decreased by only 5%, and the $V_{TH}$ was shifted by less than 0.1 V. In this long-term stability test, we rarely found a significant difference between the stand-alone and stacked devices or between the single-gate and dual-gate transistors (Supplementary Fig. 7). Further reliability tests under temperature and humidity stress conditions were conducted. Both types of transistors were heated from 30 to 150 °C and cooled down to 30 °C through the consecutive five thermal stress steps and their transfer characteristics were measured at the end of every step (Fig. 8d). The printed transistors exhibited severe performance degradation only under the thermal stress of 150 °C. When exposed to a high humidity environment (90% rh, 30 °C) for 1 day, they rarely exhibited significant changes in the transfer characteristics (Fig. 8e).

## Discussion

The 3D monolithic integration is a promising strategy for achieving technology scaling while enhancing device performance in flexible printed transistors. This study proved the feasibility of the continuous stacking of printed complementary transistors by demonstrating both 'p-type-on-n-type' and 'n-type-on-p-type' fabrication manners. We anticipate that this approach will open up possibilities for the design and production of innovative flexible printed ICs for internet of everything, wearable healthcare monitoring and smart packaging, where high device performance and the integration of hundreds of transistors into a limited plastic sheet are essential requirements. We also believe that the 3D monolithic integration strategy can extend to other printable functional materials, such as carbon nanotubes, oxide semiconductors, and 2D materials.

## Methods

**Material preparation.** All the organic transistors used in this work were fabricated on 125 μm-thick polyethylene naphthalate (Teonex PEN, DuPont) films. A hydrocarbon-based Ag-nanoparticle ink containing 55 wt% Ag nanoparticles with an average diameter of 7 nm in tetradecane (Nanopaste NPS-JL, Harima Chemicals, Inc.) was used as a conductive metal ink. The Ag nanoparticles have a relatively low sintering temperature of 120 °C, which is critical for the use of the flexible PEN substrate, whose glass transition temperature is 122 °C. To modify the work function of the printed Ag contact electrodes by self-assembled monolayer (SAM) treatment, 4-methylbenzenethiol (4-MBT) and pentafluorobenzenethiol (PFBT) were prepared in 10 mM and 30 mM solutions, respectively, using isopropanol. For the n-type semiconductor ink, an electron-transporting small-molecule (TU-3, Future Ink Corporation) was prepared in a 0.045 wt% solution using 1-methylnaphthalene (TCI Co., Ltd.) as the solvent. For the p-type semiconductor ink, a hole-transporting small molecule, dithieno[2,3-d;2',3'-d']benzo[1,2-b;4,5-b']dithiophene (DTBDT-C$_6$, Tosoh Corporation) and polystyrene (PS) were dissolved in mesitylene at concentrations of 0.6 and 0.2 wt%, respectively. Observation with polarized microscopy showed that the printed n-type semiconductor TU-3 pattern has a number of small crystal grains, while the PS-blended p-type semiconductor DTBDT-C$_6$ pattern shows a highly crystalline morphology. A hydrophobic fluoropolymer (Teflon AF1600, DuPont) prepared in a 1 wt% solution using perfluorotributylamine (Fluorinert FC-43, 3 M) as the solvent was used for surface energy modification to precisely define the semiconductor area. All the semiconductor and metal inks were filtered by using polytetrafluoroethylene

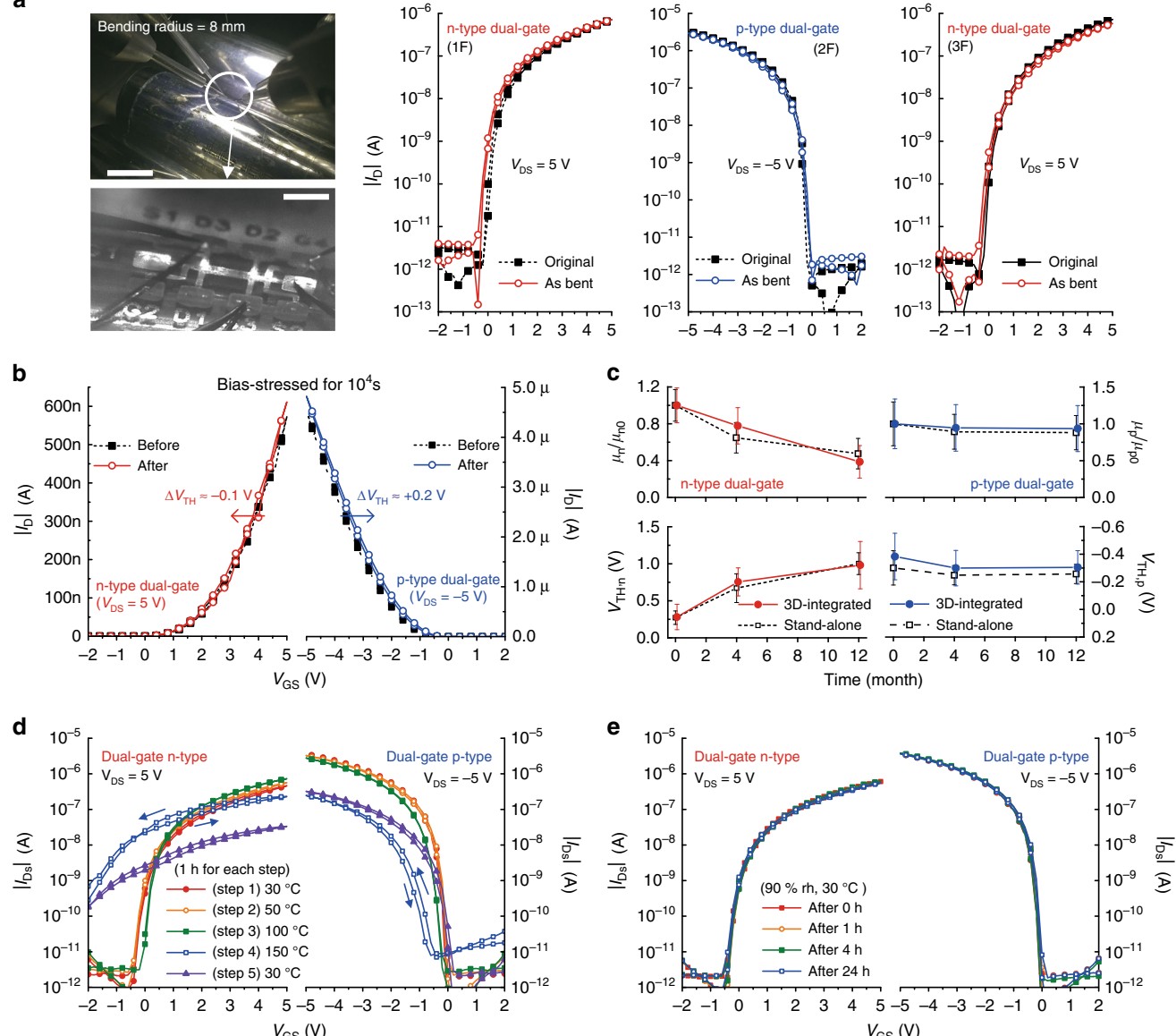

**Fig. 8** Mechanical, electrical, long-term, and environmental reliabilities **a** Images of the stacked 3-T device under bending test (white dotted area is magnified) and its saturation transfer characteristics as bent on a curved surface (scale bars in the top and bottom images are 1 cm and 1 mm, respectively). **b** Shift of the saturation transfer characteristics of the printed stand-alone complementary transistors after electrical bias stress for 10,000 s (at $V_{GS} = V_{DS} = 5$ V for the n-type and −5 V for the p-type). **c** Year-long observation on the carrier mobilities and threshold voltages of the stand-alone and stacked 2-T devices ($\mu_n$, $\mu_p$: carrier mobility, $\mu_{n0}$, $\mu_{p0}$: initial carrier mobility, $V_{TH,n}$, $V_{TH,p}$: threshold voltage, error bars represent standard deviation). **d** Shift of the saturation transfer characteristics of the complementary dual-gate transistors under consecutive five thermal stress steps. **e** Shift of the saturation transfer characteristics of the complementary dual-gate transistors under a humid environment (90% rh, 30 °C) for 24 h (rh: relative humidity)

membrane filters with a 0.45 µm pore size. For the conformal deposition of dielectric layers, which is critical for high-yield device fabrication processes, chemical vapour deposition (CVD) of a poly(p-xylylene) derivative (Parylene diX-SR, KISCO Ltd.) was used.

**Device fabrication**. For the fabrication of n-type dual-gate organic transistors, a 1 µm-thick Parylene film was formed on a 125 µm-thick PEN substrate surface by CVD to provide a controlled surface condition in the following inkjet printing step. On the Parylene-coated surface, bottom-gate electrodes were inkjet-printed using an Ag-nanoparticle ink and a drop-on-demand inkjet printer. Then, a bottom-gate dielectric Parylene film was conformally deposited to a thickness of 180 nm by the CVD process. Pairs of Ag-nanoparticle S/D parallel lines were inkjet-printed on the Parylene dielectric layer. Using a dispenser, a hydrophobic fluoropolymer, Teflon, was printed along the rectangular outlines of the transistor active areas to store OSC ink inside. For the work function modification of the printed S/D electrodes by using SAM treatment, the sample was dipped into 4-MBT solution and then

rinsed. An n-type semiconductor TU-3 ink was printed by a dispenser to fill the inside of the Teflon bank area. Again, a 180 nm-thick Parylene film was conformally formed as a top-side dielectric layer. The n-type transistor fabrication is finished by inkjet-printing top-gate electrodes. To fabricate the p-type transistors, bottom-gate electrodes of the p-type transistors were printed on the same surface. Then, a 360 nm-thick bottom-gate dielectric Parylene film, Ag S/D electrodes, Teflon bank lines, a PFBT SAM, p-type OSC DTBDT-C$_6$ patterns, a 360 nm-thick top-gate dielectric Parylene film, and top-gate electrodes were formed sequentially using the same fabrication processes. The stand-alone p-type organic transistors were not directly stacked on the n-type organic transistors but instead were fabricated next to the n-type transistors.

**Inkjet printing of Ag nanoparticles**. A drop-on-demand inkjet printer (DMP 2831, Fujifilm Dimatix) was used to print different types of Ag-nanoparticle electrodes, such as source/drain, gate, and routing interconnection. The inkjet printer was equipped with a cartridge having 16 piezo-driven nozzles for ejecting

10pl volume droplets (orifice diameter = 21.5 μm), and only a single nozzle was used for accurate and stable printing. The nozzle temperature was set at 40 °C. The positioning repeatability of the printer (±25 μm) indicates pattern-to-pattern alignment resolution in the stacking process of the functional layers. In this work, up to seven metal layers were constructed for 3D-integrated complementary organic transistors, and the layout of each layer was designed considering the maximum positioning error in order to not go beyond the alignment margin.

**Device characterization**. All the static (or DC) characteristic measurements of the printed devices were conducted by using a semiconductor parameter analyser (Keithley, 4200-SCS). The output of the 7-stage ring oscillator was measured by an oscilloscope (Tektronix, DPO2024B). The carrier mobility values of this work were extracted by fitting $I_D^{1/2}–V_{GS}$ curves in saturation regime with the following device parameters: dielectric constant, dielectric thickness, and $W/L$. A stylus profiler (Bruker, Dektak XT) was used to measure the thicknesses and roughness of thin films. The dielectric constant of the Parylene dielectric (4.1) was measured with an LCR meter (NF Corporation, ZM2376) at the lowest possible frequency (100 Hz) to rule out carrier mobility overestimation due to slow polarization.

## Data availability

The data that support the findings of this study are available from J.K. and S.J. on reasonable request.

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

## Acknowledgements

We thank Dr. H. Matsui for thoughtful discussion on the preparation of semiconducting inks. We also thank Tosoh Corporation for supplying DTBDT-C$_6$. This work was supported by grants (Code No. 2012M3A6A5055728 and 2015M3A6A5072945) from the Center for Advanced Soft Electronics under the Global Frontier Research Program of the Ministry of Science and ICT of South Korea, by the IT Consilience Creative Program (IITP-2018-2011-1-00783) supervised by IITP (Institute for Information & Communications Technology Promotion).

## Author contributions

J.K. and S.J. conceived the project. J.K. performed most experiments and Y.T. performed some of device environmental tests. Y.T. and R.S. optimized the semiconductor inks for printing. S.T., K.C. and S.J. supervised the project. J.K., S.T., K.C. and S.J. analyzed and interpreted the data, and wrote the manuscript.

## Additional information

**Competing interests:** The authors declare no competing interests.

