## [Peer Review File · Nature Communications]

Reviewers' comments:

Reviewer #1 (Remarks to the Author):

The manuscript entitled, "3D monolithic integration in flexible printed organic transistors" by Kwon *et al*. comprehensively introduces the fabrication, characterization and application of dual-gate complementary organic transistors. The application of 3D monolithic integration to vertically stack printed organic transistors is demonstrated, and applications including inverters and universal logic gate are presented. Lastly, the integrated flexible digital circuits display good stability under mechanical, electrical and environmental stresses. However, this reviewer sees no significant advance for the presented work compared to similar published work by the authors, particularly reference [20]. The reviewer suggests major revisions before acceptance of the manuscript, with the following comments to be addressed:

1. Optical images of the printed semiconductor layer are presented in the manuscript. However, a further characterization of the surface roughness is needed since the authors mentioned that the printing process produced a rough surface morphology in the semiconductor layers and Ag electrodes. Rough surfaces in printed electronics may present large barriers for carriers transporting across the interfaces. The authors might consider ink tuning methods to reduce the surface roughness. In addition, is there any reason for changing inkjet printing (as in Reference [20]) to extrusion-based printing in the deposition of the semiconducting layer?

2. As displayed in Fig. 3, carrier mobility of the n-type transistor increases with the channel length while p-type transistor does not show this trend. The authors needed to explain this phenomena and design further experiments for verification, since mobility should be independent of device geometry. Optimization of the organic transistor might benefit from such an investigation.

3. To enhance the repeatability of this work, the preservation conditions, e.g., vacuum/inert environment, temperature, humidity *etc*., for the Reliability Test should be provided.

4. The demonstrated dielectric layer was deposited by chemical vapor deposition (CVD) which requires a laser to drill the via-holes and transfer of the samples among different platforms to finish the fabrication. The authors might consider making the fabrication process more compact by replacing the CVD deposition of dielectric polymer with proper printing methods, e.g., extrusion-based printing and inkjet printing, within the tolerance of alignment and insulation requirement.

5. Minor problems related to the figures:

- The labels of electrodes "S2" and "G4" in Figure 2a are not consistent, which causes some confusion in tracking the printing orders of electrodes, especially in layer M4;
- The figure label for Fig. 3e is wrong;
- Same problem for Supplementary Fig. 9. Labels in the figure are not consistent with the description in the caption.

Reviewer #2 (Remarks to the Author):

The authors reported a 3D integration approach to achieve technology scaling in printed transistor density. They demonstrate the scalable 3D integration of dual-gate organic transistors on plastic foil by printing with high yield, uniformity and year-long stability, which are impressive.

Using printed approach to realized wrapped dual gates transistor to overcome common problems of sub-threshold swing and leakage, onoff ratio etc., is also a plus.

It's a well written paper. My only concern is that the transistor density , although claimed to be the highest, is 60/cm², which is not really microelectronics. I suggest the authors to extend discussion on the realistic limit of this technology, and the more specific application areas, which will be beneficial for broad audience.

Reviewer #3 (Remarks to the Author):

The authors discuss in this work a scalable 3D integration route of TFTs on plastic foil by printing techniques. The content of this paper is valuable for the field, the results are of impressive quality. I do have some minor comments/suggestions for this manuscript.

1. The discussion on logic gates is mainly focused on static behaviour. Could you provide timing information of the logic gates to discuss the dynamic behaviour? A comparison between 2-D and 3-D stacked devices on dynamic performance would be a great add-on to the paper.
2. Related to the static behaviour: would it be possible to comment on the currents, such that the readers can be informed on static power consumption? Is it possible to add the noise margin (of e.g. the NOT gate)?
3. Fig. 3: can you please provide information on the W/L of the devices? (e) is wrongly labeled in the figure.
4. Can you comment why the top gate n-TFT performs better compared to bottom gate?
5. I would suggest to improve the quality of Fig. 4, to fit Nature standards. Especially Fig. 4 d, there is a lot of information in some of the graphs. Even some arrows are pointing nowhere.
6. Can you improve the quality of Fig. 5, to make the texts more readable.
7. Line 139: The mobility is a material property, contact resistance should not influence the mobility, of course it would influence the extracted mobility.
8. Line 185-187: were the devices stored in air?

Reviewer #1

The manuscript entitled, “3D monolithic integration in flexible printed organic transistors” by Kwon *et al.* comprehensively introduces the fabrication, characterization and application of dual-gate complementary organic transistors. The application of 3D monolithic integration to vertically stack printed organic transistors is demonstrated, and applications including inverters and universal logic gate are presented. Lastly, the integrated flexible digital circuits display good stability under mechanical, electrical and environmental stresses.

1. However, this reviewer sees no significant advance for the presented work compared to similar published work by the authors, particularly reference [20].

As the reviewer pointed out, this study builds on our 2016 paper in ACS Nano¹, which reports the vertical stacking of two complementary organic transistors. In this study, we address two key challenges to the realization of the emerging applications of flexible printed organic transistors in printed electronics as follows: (We had mentioned the same in our *Cover Letter to Editor*)

- (i) This work for the first time reports a strategy to continuously downsize printed complementary organic TFTs through 3D monolithic integration.. The direct printing of thin-film transistors urgently requires technology scaling in transistor density, analogous to Moore’s law, which has been driven by lithography for the last 50 years. Unlike previous studies on stacking thin film transistors²⁵⁻²⁷, we have shown the continuous stacking of complementary transistors by demonstrating both ‘p-type-on-n-type’ and ‘n-type-on-p-type’ fabrication processes. Furthermore, we demonstrate the 3D integration of dual-gate organic transistors on plastic foil with a record density of 60 printed transistors per square centimetre. The high yield, uniformity, and year-long

stability of the 3D-printed transistors prove that the footprint of the printed device can be continually reduced to create a scaling law for printed and flexible electronics.

- (ii) We have adopted a dual-gate configuration in the 3D transistor-on-transistor structure. This configuration yields advantages that help overcome the chronic problems of conventional single-gate organic transistors. Through effective control of charge transport within the bulk of the semiconductor, this device structure provides a fully depleted semiconductor bulk, leading to exceptional electrical properties such as subthreshold swing, on/off ratio and transconductance, compared to those of the printed single-gate TFTs (Fig. 3). In addition, a dual-gate configuration has a structural advantage for gate-sharing TFT stacking process.

Based on the strategies mentioned above, we have demonstrated a 3D universal logic gate (NAND) by interconnecting three dual-gate transistors for the first time. We then introduce the 3D NAND array that can be programmed by inkjet-printing. We believe this is a new and unique route to design and produce digital circuitry essential for emerging applications, which has not ever been suggested nor demonstrated in any other transistor technologies.

To clarify the advances, we have added a paragraph at the end of the introduction section as follows:

“We previously introduced a three-dimensional (3D) organic transistor structure with a shared gate joining two complementary transistors using drop-on-demand inkjet printing²⁰. The study reported an array of the vertically stacked complementary transistors and their logic gates. Building on the approach, here we propose the 3D monolithic integration of flexible printed organic transistors to realize technology scaling and performance enhancement. In addition, we adopt a dual-gate configuration in the 3D transistor-on-transistor structure for the improvement of their electrical characteristics such as subthreshold swing, transconductance, and drain current on-off ratio. The 3D monolithic integration of dual-gate organic transistors was successfully implemented on a plastic foil with a record density of 60 printed transistors per square centimetre. These transistors exhibited high yield, uniformity, and year-long stability, which prove that this technology is extendable to large-scale printed electronics. By interconnecting those 3D-integrated dual-gate transistors, we finally propose a 3D universal logic gate and its array as a new facile route to design printed digital circuitries that are essential for emerging flexible electronics applications.”

2. Optical images of the printed semiconductor layer are presented in the manuscript. However, a further characterization of the surface roughness is needed since the authors mentioned that the printing process produced a rough surface morphology in the semiconductor layers and Ag electrodes.

To characterize the roughness of the printed surfaces, we have added AFM topology data of a Parylene-coated PEN substrate and printed complementary OSCs to **Supplementary Fig. 2**. The Parylene-coated PEN substrate had R_q and R_a of 3.81 and 2.71 nm, respectively. As printed on the Parylene-coated PEN film, the polycrystalline n-type OSC TU-3 exhibited higher roughness values ($R_q = 22.7$ nm, $R_a = 18.3$ nm) than those of the highly ordered p-type OSC DTBDT-C₆. The roughness of the printed DTBDT-C₆ film ($R_q = 3.26$ nm, $R_a = 2.49$ nm) was not markedly different from that of the substrate. Those roughness results are well matched with the polarized microscopy data of **Supplementary Fig. 1c and d** showing the crystallinity of each OSC material.

The morphological characteristics of Ag patterns printed using the Harima NPS-JL ink were reported in the previous study¹. The carbon-based Ag ink produced coffee-ring profiles in dot and line patterns when it was inkjet-printed. Here we adopted the same printing approach and material for Ag electrodes, but we used a different substrate (Parylene-coated PEN). **Redacted**

Redacted

3. Rough surfaces in printed electronics may present large barriers for carriers transporting across the interfaces.

As seen in **Fig. 3**, the thin-film transistors (TFTs) printed on different floors showed little difference in their performance parameters (e.g. carrier mobility and threshold voltage). It tells that surface roughness of the bottom stacked layers would not significantly affect the performance of the printed TFTs fabricated on the top.

One possible reason of the irrelevance between the surface roughness and the carrier mobility is the bulk accumulation in the dual-gate TFTs. It has been reported that a dual-gate configuration enhances charge accumulation in semiconductor bulk so that entire charge transport in a dual-gate TFT can be less affected by the interface condition than in a single-gate TFT^{2,3}.

4. The authors might consider ink tuning methods to reduce the surface roughness.

Through preliminary experiments, we optimized the ink formulation (concentration and solvent) for each OSC material. For example, the p-type OSC DTBDT-C₆ was initially prepared in toluene as used in the previous planar printed TFT fabrication⁴. However, cracks and rough morphology were observed in the printed DTBDT-C₆ patterns, as seen in **Figure R2**, leading to low yield in our vertically stacked TFTs. To form well-ordered DTBDT-C₆ patterns during the drying process, we used different solvents with lower polarity and higher boiling point. As shown in **Supplementary Figures 1d**, we found that the mesitylene-based ink with the optimized DTBDT-C₆:PS weight ratio of 3:1 resulted in the formation of highly crystalline films over the patterned area without unwanted cracks.

Figure R2 Polarized microscopy of the p-type OSC film printed on a Parylene-coated PEN substrate using the toluene-based DTBBDT-C₆ ink. Cracks and irregular morphology are observed. The white dotted areas are magnified.

For the n-type OSC TU-3, we chose the solvent for optimizing n-type TFT performance. Unlike the DTBBDT-C₆, the TU-3 exhibited highly polycrystalline and rough morphologies in printed films even when we used a high boiling point solvent. However, even though the overall roughness of the stacked films increased due to the TU-3, it did not significantly affect the performance of our stacked TFTs. Moreover, the 3D-integrated TFTs exhibited high device yield close to 100 % and exceptional uniformity.

5. In addition, is there any reason for changing inkjet printing (as in Reference [20]) to extrusion-based printing in the deposition of the semiconducting layer?

In our previous studies on manufacturing printed TFTs^{1,5-7} we have found that the dispensing method can be more effective to form high-quality small-molecule OSC films than the inkjet printing. The dispensing printing can eject the large volume of liquid at once, making it easy to fill the inside of a bank with OSC ink. The active area can be precisely guided by the hydrophobic bank lines, and the small molecules can be maximally crystallized as the ink is slowly dried.

To optimise the performance of our printed complementary TFTs, we carefully decided the dispensing volume of OSC ink to be stored inside a bank area. As seen in **Figure R3a**, the volume of printed ink can be controlled by adjusting dispensing air pressure. When the ink volume is too small, the OSC film cannot cover the whole active channel area, and then the printed TFTs would have low device performance. For example, in the case of the n-type TU-3 ink, the extracted carrier mobility of the printed transistors is

Figure R3 a Microscopic images of four printed TU-3 dual-gate transistors with different OSC dispensing volume conditions. **b** Saturation transfer characteristics (normalized $|I_D|$ vs. V_{GS}) of the four n-type dual-gate printed transistors.

saturated to 0.07–0.08 $\text{cm}^2\text{V}^{-1}\text{s}^{-1}$ at a dispensing pressure larger than 6 kPa as seen in **Figure R3b**.

- As displayed in Fig. 3, carrier mobility of the n-type transistor increases with the channel length while p-type transistor does not show this trend. The authors needed to explain this phenomena and design further experiments for verification, since mobility should be independent of device geometry. Optimization of the organic transistor might benefit from such an investigation.

In the paragraph related to Fig. 3e, we intended to explain (i) why the extracted carrier mobility values of the stacked n-type transistors were different in Fig. 3d, and (ii) why the top gate performed better than the bottom gate in the n-type dual-gate transistors in Supplementary Fig. 6a. We can attribute those two phenomena to the high contact resistance of the n-type transistors. The high contact resistance can severely limit the performance of organic transistors which are downscaled with short L ⁸. Since the mobility extraction using a slope of $I_{DS}^{1/2}$ – V_{GS} transfer characteristics in saturation regime cannot eliminate the influence of contact resistance, we have observed lower mobility values in the printed devices with shorter L .

To help readers better understand, we have added results on contact resistance that were measured in both types of dual-gate transistors by using TLM. As can be presented in **Supplementary Fig. 5a**, the n-type devices showed ten times higher contact resistance than that of the p-type devices. To answer (i) and (ii), we revised the paragraph as follows:

“We observed that the extracted carrier mobilities of the n-type transistors decreased with shortening the channel length, while those of the p-type devices remained unchanged (Fig. 3e)²⁹. We hypothesized that the behaviour of the n-type transistors was dominated by charge injection due to high contact resistance. To examine this, we measured and analysed the contact resistance of the transistors using the transmission line method. As can be seen in Supplementary Fig. 5a, the contact resistance of the n-type dual-gate transistors was 21.4 kΩ·cm at $V_{GS} = 5$ V which was ten times higher than that of the p-type dual-gate transistors (2.2 kΩ·cm at $V_{GS} = -5$ V). The n-type transistors showed the functional dependence of the contact resistance with the gate bias, which resulted in the improvement of the extracted mobility (Supplementary Fig. 5b). In addition, the n-type dual-gate devices exhibited highly asymmetrical performance between top-gate and bottom-gate operation modes (Supplementary Fig. 6a). This asymmetrical operation can be attributed to the difference in charge injection efficiency between the staggered top-gate structure and the bottom-gate structure in the dual-gate transistors. In contrast, the carrier mobilities of the p-type transistors were weakly dependent on the gate bias due to sufficiently low contact resistance, which enabled remarkably symmetrical operation as shown in Supplementary Fig. 6b.”

7. To enhance the repeatability of this work, the preservation conditions, e.g., vacuum/inert environment, temperature, humidity *etc.*, for the Reliability Test should be provided.

We believe that our results on the printed TFTs are reliable and repeatable because the measurement data were acquired through the multiple rounds of experiments as seen in Fig. 5, Table 2, and Supplementary Fig. 4. As you requested, we have additionally conducted the short-term reliability tests of the printed dual-gate TFTs in different temperature and humidity conditions. The test results are added to **Supplementary Fig. 10b and c** and the manuscript is revised as follow:

“Results on further reliability tests under temperature and humidity stress conditions are provided in

Supplementary Fig. 10b and c.”

8. The demonstrated dielectric layer was deposited by chemical vapor deposition (CVD) which requires a laser to drill the via-holes and transfer of the samples among different platforms to finish the fabrication. The authors might consider making the fabrication process more compact by replacing the CVD deposition of dielectric polymer with proper printing methods, e.g., extrusion-based printing and inkjet printing, within the tolerance of alignment and insulation requirement.

The 3D-integrated devices of this work were fully printed except the Parylene dielectric layers which were deposited by CVD. A key challenge on the 3D printing fabrication of the TFTs and circuits was not to damage the underlying functional films during solution processes. Recently, we have studied solution-processable polymer dielectrics for 3D-integrated organic TFT structures in the previous studies^{9,10}, and have obtained reasonable device yield and performance. However, we could not achieve 100% yield with those solution-processable dielectric materials, which is essential for the reliable operation of integrated circuits. Although it is not printable, Parylene provides conformal and pinhole-free coating with excellent solvent resistance and good electrical/physical insulation properties, which enables us to develop 3D-printed ICs on plastic with exceptionally high yield and uniformity.

Minor problems related to the figures:

9. The labels of electrodes “S2” and “G4” in Figure 2a are not consistent, which causes some confusion in tracking the printing orders of electrodes, especially in layer M4.

We have corrected the error of **Fig. 2**. As the reviewer said, the S2/D2 electrodes belong to the layer M4 and the G4 electrodes is on the top of the other layers.

10. The figure label for Fig. 3e is wrong.

We have corrected the labelling error of **Fig. 3e**.

11. Same problem for Supplementary Fig. 9. Labels in the figure are not consistent with the description in the caption.

We have corrected the labelling error of **Supplementary Figure 9** to be consistent with the figure description.

Reviewer #2

The authors reported a 3D integration approach to achieve technology scaling in printed transistor density. They demonstrate the scalable 3D integration of dual-gate organic transistors on plastic foil by printing with high yield, uniformity and year-long stability, which are impressive. Using printed approach to realized wrapped dual gates transistor to overcome common problems of sub-threshold swing and leakage, on/off ratio etc., is also a plus. It is a well-written paper.

Thank you.

1. My only concern is that the transistor density, although claimed to be the highest, is 60 cm^{-2} , which is not really microelectronics. I suggest the authors to extend discussion on the realistic limit of this technology, and the more specific application areas, which will be beneficial for broad audience.

To discuss the limit and more specific application of this technology, we have added sentences to the manuscript as follows:

“Based on this technology, we could fabricate up to ≈ 2700 programmable transistors on the size of a standard credit card ($85.60 \times 53.98 \text{ mm}$), which is compatible with the transistor count of the first commercial 4-bit microprocessor.”

“We anticipate that this approach opens up possibilities for the design and production of innovative flexible printed ICs for internet of everything, wearable healthcare monitoring and smart packaging, where high device performance and the integration of hundreds of transistors into a limited plastic sheet are essential requirements.”

Reviewer #3

The authors discuss in this work a scalable 3D integration route of TFTs on plastic foil by printing techniques.

The content of this paper is valuable for the field, the results are of impressive quality.

Thank you.

I do have some minor comments/suggestions for this manuscript.

1. The discussion on logic gates is mainly focused on static behaviour. Could you provide timing information of the logic gates to discuss the dynamic behaviour? A comparison between 2-D and 3-D stacked devices on dynamic performance would be a great add-on to the paper.

We observed the dynamic characteristics of the 3D devices by using a 7-stage ring oscillator. We have added the schematic and ring oscillation operation to **Supplementary Fig. 7c**. We have also added information on the gate delay to the 3D inverter description part in the manuscript as follows:

“Furthermore, we explored the dynamic operation of the 3D-integrated complementary dual-gate transistors by fabricating a 7-stage ring oscillator. The circuit consists of seven inverters for the ring oscillation operation and two inverters for the voltage buffering. The ring oscillator operated with the supply voltage V_{DD} varying between 1 and 15 V. The gate delay was 13 ns at $V_{DD} = 1$ V and 340 μ s at $V_{DD} = 15$ V. Supplementary Fig. 7c depicts the schematic circuit and the oscillation operation at 48.5 Hz when the V_{DD} is 5 V.”

As the reviewer mentioned, the dynamic behavior comparison between the 2D- and 3D-integrated devices is important for better understanding of our 3D monolithic integration process. For this, we are currently working on a capacitance model of the proposed 3D-integrated complementary TFTs by focusing on the difference with the conventional 2D planar TFTs. In our future paper, we will report new design issues of the 3D-integrated TFTs considering their dynamic characteristics.

2. Related to the static behaviour: would it be possible to comment on the currents, such that the readers can be informed on static power consumption? Is it possible to add the noise margin (of e.g. the NOT gate)?

We have added the static current graph of the inverter operation to Supplementary Fig. 7b, and static noise margin value of the 3D NAND gate to Fig. 4b.

3. Fig. 3: can you please provide information on the W/L of the devices?

We have added all the channel geometries, extracted carrier mobility in saturation regime, and threshold voltage in **Supplementary Table 2**.

4. (e) is wrongly labeled in the figure.

We have corrected the labelling error in **Fig. 3**.

5. Can you comment why the top gate n-TFT performs better compared to bottom gate?

As is well known, a staggered TFT configuration, which has wider charge injection region, generally performs better than a coplanar TFT configuration¹¹. The lower the charge injection efficiency between OSC and contact electrodes leads to the greater the discrepancy in performance between the two configurations. Because the n-type transistors have contact resistance ten times higher than that of the p-type transistors (**Supplementary Fig. 5a**), this phenomenon becomes more prominent.

In this work, as the printed Ag electrodes are always deposited under the OSC films, the top- and bottom-gate TFTs have no choice but to be constructed in staggered and coplanar structures, respectively (see **Supplementary Fig. 3**). Therefore, it seems to be reasonable that the single-gate n-type TFTs performed better in a top-gate staggered configuration than in a bottom-gate coplanar configuration. In the same way, we can understand why the top-gate operation (staggered gate-contact) performed better than the bottom-gate operation (coplanar gate-contact) in the dual-gate n-type TFTs (**Supplementary Fig. 6a**).

We revised the manuscript to explain this as follows:

“In addition, the n-type dual-gate devices exhibited highly asymmetrical performance between top-gate and bottom-gate operation modes (Supplementary Fig. 6a). This asymmetrical operation can be

attributed to the difference in charge injection efficiency between the staggered top-gate structure and the bottom-gate structure in the dual-gate transistors. In contrast, the carrier mobilities of the p-type transistors were weakly dependent on the gate bias due to sufficiently low contact resistance, which enabled remarkably symmetrical operation as shown in Supplementary Fig. 6b.”

6. I would suggest to improve the quality of Fig. 4, to fit Nature standards. Especially Fig. 4 d, there is a lot of information in some of the graphs. Even some arrows are pointing nowhere. Can you improve the quality of Fig. 5, to make the texts more readable.

We have amended Fig 4 as requested, and have revised the texts and caption of Fig. 5.

7. Line 139: The mobility is a material property, contact resistance should not influence the mobility, of course it would influence the extracted mobility.

In the paragraph related to Fig. 3e, we intended to explain (i) why the extracted carrier mobility values of the stacked n-type transistors were different in Fig. 3d, and (ii) why the top gates performed better than the bottom gates in the n-type dual-gate transistors. We can attribute those two phenomena to the high contact resistance of the n-type transistors. The high contact resistance can severely limit the performance of organic transistors which are downscaled with short L ⁸. Since the mobility extraction using a slope of $I_{DS}^{1/2}$ - V_{GS} transfer characteristics in saturation regime cannot eliminate the influence of contact resistance, we have observed lower mobility values in the printed devices with shorter L .

To help readers better understand, we have added results on contact resistance that were measured in both types of dual-gate transistors by using TLM. As can be presented in **Supplementary Fig. 5a**, the n-type devices showed ten times higher contact resistance than that of the p-type devices. To answer (i) and (ii), we revised the paragraph as follows:

“We observed that the extracted carrier mobilities of the n-type transistors decreased with shortening the channel length, while those of the p-type devices remained unchanged (Fig. 3e)²⁹. We hypothesized that the behaviour of the n-type transistors was dominated by charge injection due to high contact resistance. To examine this, we measured and analysed the contact resistance of the

transistors using the transmission line method. As can be seen in Supplementary Fig. 5a, the contact resistance of the n-type dual-gate transistors was 21.4 k Ω ·cm at $V_{GS} = 5$ V which was ten times higher than that of the p-type dual-gate transistors (2.2 k Ω ·cm at $V_{GS} = -5$ V). The n-type transistors showed the functional dependence of the contact resistance with the gate bias, which resulted in the improvement of the extracted mobility (Supplementary Fig. 5b). In addition, the n-type dual-gate devices exhibited highly asymmetrical performance between top-gate and bottom-gate operation modes (Supplementary Fig. 6a). This asymmetrical operation can be attributed to the difference in charge injection efficiency between the staggered top-gate structure and the bottom-gate structure in the dual-gate transistors. In contrast, the carrier mobilities of the p-type transistors were weakly dependent on the gate bias due to sufficiently low contact resistance, which enabled remarkably symmetrical operation as shown in Supplementary Fig. 6b.”

8. Line 185-187: were the devices stored in air?

The samples were stored in a dry ambient environment. We have revised the sentence as follows:

“Remarkably, the electrical operations of the printed complementary organic transistors were maintained for one year with acceptable performance changes when they were stored in a dry ambient environment (Fig. 5c).”

Reference

1. Takeda, Y. *et al.* Integrated circuits using fully solution-processed organic TFT devices with printed silver electrodes. *Org. Electron.* **14**, 3362–3370 (2013).
2. Mativenga, M., An, S. & Jang, J. Bulk Accumulation a-IGZO TFT for High Current and Turn-On Voltage Uniformity. *IEEE Electron Device Lett.* **34**, 1533–1535 (2013).
3. Li, X., Geng, D., Mativenga, M., Chen, Y. & Jang, J. Effect of bulk-accumulation on switching speed of dual-gate a-IGZO TFT-based circuits. *IEEE Electron Device Lett.* **35**, 1242–1244 (2014).
4. Shiwaku, R. *et al.* Printed Organic Inverter Circuits with Ultralow Operating Voltages. *Adv. Electron. Mater.* **3**, 1600557 (2017).
5. Fukuda, K., Takeda, Y., Mizukami, M., Kumaki, D. & Tokito, S. Fully Solution-Processed Flexible Organic Thin Film Transistor Arrays with High Mobility and Exceptional Uniformity. *Sci. Rep.* **4**, 3947 (2014).
6. Shiwaku, R. *et al.* Printed 2 V-operating organic inverter arrays employing a small-molecule/polymer blend. *Sci. Rep.* **6**, 34723 (2016).
7. Matsui, H. *et al.* Printed 5-V organic operational amplifiers for various signal processing. *Sci. Rep.* **8**, 8980 (2018).
8. Hoppe, A. *et al.* Scaling limits of organic thin film transistors. *Org. Electron. physics, Mater. Appl.* **11**, 626–631 (2010).
9. Kwon, J., Kyung, S., Yoon, S., Kim, J.-J. & Jung, S. Solution-Processed Vertically Stacked Complementary Organic Circuits with Inkjet-Printed Routing. *Adv. Sci.* **3**, 1500439 (2016).
10. Kyung, S., Kwon, J., Kim, Y.-H. & Jung, S. Low-Temperature, Solution-Processed, 3-D Complementary Organic FETs on Flexible Substrate. *IEEE Trans. Electron Devices* **64**, (2017).
11. Kim, C. H., Bonnassieux, Y. & Horowitz, G. Fundamental Benefits of the Staggered Geometry for Organic Field-Effect Transistors. *IEEE Electron Device Lett.* **32**, 1302–1304 (2011).

REVIEWERS' COMMENTS:

Reviewer #1 (Remarks to the Author):

In the response letter and revised manuscript, including supporting information, the authors have provided supplementary data to support their argument and the questions raised by the reviewer have been sufficiently answered. This research provides a promising methodology to improve the density and functionality of TFTs in flexible ICs. It is now recommended for publication.

Reviewer #2 (Remarks to the Author):

The authors provide relatively detailed reply to the questions and concerns from the reviewers. I am satisfied with the revised version of the manuscript.

Reviewer #3 (Remarks to the Author):

I would like to thank the authors for their comprehensive rebuttal letter and their amendments to the manuscript and supplementary material. Besides some remarks below, I have no further comments.

Regarding comment # of reviewer 3, it is indeed clear that the capacitance model is needed for improved designs and their challenges for their 3D integrated TFTs. It is no problem from my side that this model is not final for this work, as the manuscript as such is already very valuable for the field.

Regarding comment #2: To my opinion, a SNM of 50 and 47% for both NAND gates cannot be correct, since the slope is not a perfect straight line and the curves are not ideal. My suggestion: please recalculate it according to the maximal equal criterion, or remove it from the graphs.

Reviewer #3

Regarding comment #2: To my opinion, a SNM of 50 and 47% for both NAND gates cannot be correct, since the slope is not a perfect straight line and the curves are not ideal. My suggestion: please recalculate it according to the maximal equal criterion or remove it from the graphs.

The SNM is the maximum length of a square fitted to the butterfly graph divided by $V_{DD}/2$ as seen in the figure below (Kwon et al. *ACS Nano* 2016). Therefore, the ideal SNM value should be 100%. To avoid confusion, we deleted the SNM information from the NAND gate transfer graphs